# Analysis of Volatile and Non-Volatile Components of Dried Chili Pepper (*Capsicum annuum* L.)

**DOI:** 10.3390/foods14050712

**Published:** 2025-02-20

**Authors:** Wenqi Li, Yuan Wang, Lijie Xing, Wensheng Song, Shiling Lu

**Affiliations:** 1College of Food Science, Shihezi University, Shihezi 832000, China; 15291649041@163.com (W.L.); lushiling_76@163.com (S.L.); 2Analysis and Testing Center, Xinjiang Academy of Agriculture and Reclamation Science, Shihezi 832000, China; 15001640887@163.com; 3Xinjiang Tianjiao Hongan Agricultural Technology Co., Ltd., Shihezi 832000, China; sws6911@126.com

**Keywords:** sensory evaluation, nutritional index, HS-GC-TOF MS

## Abstract

As an important crop in the world, dried pepper is widely used in various foods. However, the sensory quality, fruit shape index, edible index, nutrition index, and volatile components of dried pepper have not been comprehensively analyzed. This study elucidated the differences between different varieties of dried pepper and provided the basis for the selection of raw materials for different varieties of dried pepper products. The varieties with high scores in sensory evaluation were Henan new generation, Neihuang new generation, Chengdu Erjingtiao, India S17, and Honglong 12. The varieties with the highest fruit shape index, edible rate, and nutrition index were Chengdu Erjingtiao and Guizhou Erjingtiao. A total of 380 volatile organic compounds were identified by comprehensive two-dimensional gas chromatography–time-of-flight mass spectrometry with headspace sampling (HS-GC-TOF MS), including 62 alcohols, 50 aldehydes, 68 ketones, 60 hydrocarbons, 99 esters, 18 acids, and 23 other substances such as pyrazoles and ethers.

## 1. Introduction

Chili pepper (*Capsicum annuum* L.) is an annual or limited perennial herb. Chili pepper fruit can increase appetite because its skin contains capsaicin and has a spicy flavor, and the global pepper planting area covers more than 50 million acres. Chili pepper is rich in capsaicin [1], vitamin C [2], and carotenoids [3]. Because of its unique flavor characteristics and health effects [4], chili pepper is a major consumer product worldwide. Chili peppers come in many varieties and play numerous roles in the cooking process, including providing color [5], adding heat, and providing flavor [6]. Different varieties of chili peppers have different nutrition levels, colors, and fruit shape indices [7], and these indicators affect consumer acceptance. *C. chinense* Jacq is widely utilized because of its unique smell and spicy taste [8]. Measurement of these indices can provide comprehensive information on dried chili peppers and help in evaluating their quality.

Food flavor is composed of volatile and non-volatile components. Studies have shown that volatile components are the main factors affecting food flavor and that interactions between volatile components can promote complex flavors in food. However, not all volatile components in a food significantly contribute to the aroma, and only a few key volatile components contribute to the overall flavor [9]. It is necessary to evaluate food quality by measuring all aroma components. The analysis of key aroma compounds can improve the sensory quality of food [10]. Yamasaki et al.’s [11] studies have shown that pepper from a certain region of Japan can be classified on the basis of sensory evaluation and volatile component determination. However, there are no relevant studies on the quality analysis of dried chili pepper based on volatile and non-volatile components.

Sensory evaluation can analyze subtle differences that cannot be detected by instruments, which can be described with a series of sensory descriptors [12]. The commonly used instrumental analysis techniques for the identification of volatile components in food include the electronic nose, gas chromatography–mass spectrometry (GC–MS), and gas chromatography–odor–mass spectrometry [13]. In recent years, gas chromatography–time-of-flight mass spectrometry (GC–TOF MS) has been increasingly used in the detection of volatile components in food, and compared with GC–MS, GC–TOF MS can detect more subtle differences [14]. GC–TOF MS is superior to ordinary MS in terms of resolution, mass accuracy, sensitivity, scanning speed, and detection limit [15].

In this work, the non-volatile components and volatile components of dried chili pepper were identified via headspace solid-phase microextraction–gas chromatography–time-of-flight mass spectrometry (HS-SPME-GC-TOF MS). The results provide important information for the dried chili pepper processing industry.

## 2. Materials and Methods

### 2.1. Plant Material

In this study, 18 representative chili pepper (*Capsicum annuum* L.) varieties were selected. Honglong12 (XHR), Hongguan (XG), and Honglong23 (XHS) are from Xinjiang Tianjiao Hongan Agricultural Technology Co., Ltd. (Shihezi, China). Chengdu erjingtiao (CBR), Shizhuohong5 (CW), Shizhuohong3 (CSS), Xinjiang zhongjiao (CXZ), Guizhou erjingtiao (CGR), India S17 (CYS), Neihuangxinyidai (CN), Guizhou mantianxing (CGM), bell pepper (CD), new generations of small pepper (CX), and Guizhou zidantou (CG) were sourced from China’s Chengdu Farmers Market. Chongqing mantianxing (CM), Henan neihuangxinyidai (CHN), Henan xinyidai (CHX), and fresh millet pepper (XM) were sourced from China’s Chongqing Farmers Market. Ethanol (≥95%), anthranone, metaphosphoric acid (40–50%), glucose (≥99%), ammonium molybdate tetrahydrate (≥99%), L-ascorbic acid (≥99%), Coomassie bright blue G-250, and bovine serum protein (pH = 7.0) were purchased from Shanghai Anpu Experimental Technology Co., Ltd. (Shanghai, China).

### 2.2. Preparation of Chili Pepper

After all the chili peppers were dried by hot air at same temperatures, they were removed from the stalks, crushed with a grinder, and passed through a 40-mesh sieve. The chili pepper powder was stored in a refrigerator at −20 °C.

### 2.3. Sensory Evaluation

Dried chili peppers are food products that have been evaluated as safe to eat and do not require ethical permission for sensory evaluation. Eleven people were selected, and all panelists were confirmed to have no spicy addiction or rejection, with an age range of 23–27 years and a typical taste sensitivity that meets the requirements of ISO8586 (2012) [16]. All group members were not allowed to smoke, eat, or drink water for one hour prior to the evaluation and were not exposed to chili-related flavorings for 90 min prior to the test. The panelists did not use cosmetics or other odors during the evaluation period. Participants provided informed consent by stating, “I understand that my responses are confidential and I agree to participate in this sensory evaluation”, which requires an affirmative response in order to enter the sensory evaluation. They could withdraw from the sensory assessment at any time without giving any reason. The evaluation room had low noise levels, constant temperature and humidity, air circulation, and no smell of impurity, and the indoor light was suitable. C. Li et al. [17] classified the odor descriptors of dried peppers according to previous research before sensory testing. Color, odor, and taste were selected as sensory descriptors, a sensory rating table was developed, and the sensory evaluation team was asked to score the chili pepper samples.

### 2.4. Edible Rate Measurement

An electronic balance was used to determine the whole fruit weight of the chili pepper fruits, and the part remaining after the stem and seeds were removed was weighed again. Each variety was measured 10 times in parallel, and the edible rate of fruit was calculated from the recorded data.

### 2.5. Fruit Shape Index Determination

The largest transverse diameter and longitudinal diameter of each fruit were measured with Vernier calipers. Ten parallel measurements were made for each variety to calculate the fruit shape index. The folding and shape of the fruit surface were observed and recorded.Fruit shape index = W/H × 100%(1)

W—Fruit length, cm;H—Fruit width, cm.

### 2.6. Ash Determination

A total of 3 g dried chili pepper powder was weighed into a crucible. The crucible was placed on an electric heating plate, covered in half, and then heated carefully to ensure that the sample was completely carbonized under smokeless conditions with ventilation. The crucible was placed into a high-temperature furnace; the temperature was raised to 900 °C for 1 h, then cooled to 200 °C, and finally, the crucible was put into a dryer to cool for 30 min.

The ash content was calculated according to the following formula:x = (m_1_ − m_2_)/[(m_3_ − m_2_) × W] × 100%(2)

m_1_—Mass of crucible and ash, g;m_2_—Mass of crucible, g;m_3_—Mass of crucible and sample, g;W—Sample dry matter content (mass fraction), %.

### 2.7. Protein Determination

The protein content of the dried chili pepper was determined via the Coomassie bright blue method. A total of 0.06 g of treated dried chili pepper was placed in a 15 mL centrifuge tube, 9 mL of distilled water was added, the mixture was centrifuged at 6000 r·min^−1^ for 10 min, 1 mL of supernatant was collected, 5 mL of Coomassie blue solution was added, and the mixture was shaken well and incubated for approximately 10 min. The absorbance at 595 nm was measured, and the water-soluble protein content (µg·mL^−1^) was determined with a colorimetric dish.

### 2.8. Reduced Vitamin C Determination

The molybdenum blue colorimetric method was used to determine the reduced vitamin C content of the dried chili pepper powders. A total of 5 g of treated dried chili pepper powder was added to an appropriate amount of oxalate-EDTA solution and centrifuged at 3500 r·min^−1^ for 5 min. A total of 1 mL of supernatant was removed, and 100 µL of metaphosphoric acid–acetic acid solution, 200 µL of 5% sulfuric acid solution, and 400 µL of 5% ammonium molybdate solution were added. The absorbance was measured at 709 nm by using distilled water, shaking well, and letting stand for 15 min.

The reduced vitamin C content in the sample was calculated according to the absorbance of the liquid sample:Reduced vitamin C = [(c × v_1_)/(w × v_2_)] × 100%(3)

c—Content of reduced vitamin C in the tested sample solution, mg;v_1_—Total volume of liquid, mL;v_2_—Liquid volume used for determination, mL;w—Sample weight, g.

### 2.9. Determination of Soluble Sugars

The soluble sugar content of the dried cayenne pepper powder was determined through copper reduction iodometry. A total of 2 g of dried chili pepper powder was precisely weighed into a beaker, moistened with a small amount of distilled water, and washed into a 100 mL volumetric flask with water. The sample was then heated in a boiling water bath for 10 min. After cooling, 1 mL each of potassium ferricyanide solution and zinc sulfate solution were added. The flask was shaken well, the volume was brought up to the mark, and the sample was filtered for subsequent use. Using a pipette, 5 mL of the filtrate was transferred to a 50 mL volumetric flask. A total of 1 mL of hydrochloric acid solution was added, and the sample was heated in a boiling water bath for 10 min. After cooling, phenolphthalein indicator was added, and 0.5 mol/L of sodium hydroxide solution was added to neutralize the solution until it turned red; then, diluted hydrochloric acid was added until the red color disappeared and the volume was made constant.

The soluble sugar content was calculated via the following formula:W = [a + b(v_0_ − v_4_)] × v_1_ × v_2_/(v_2_ × v_5_ × m × 1000)(4)

v_0_—Titration volume of 0.005 mol/L sodium thiosulfate solution used for the blank solution, mL;v_1_—Total volume of the dried chili pepper filtrate, mL;v_2_—Volume of the dry chili pepper filtrate, mL;v_3_—Individual volume of the dry chili pepper filtrate, mL;v_4_—Titration volume of 0.005 mol/L sodium thiosulfate solution used for the dried chili pepper filtrate, mL;v_5_—Volume of dried chili pepper filtrate used for determination, mL;m—Mass of dried chili pepper, g;W—Soluble sugar content in the dried chili pepper powder (%).

### 2.10. Determination of Fat Content

A total of 2 g of dried chili pepper powder was precisely weighed and transferred entirely into a filter paper cartridge. The filter paper cartridge was placed in a Soxhlet extractor for 6 h. The receiving bottle was removed. When 1–2 mL of solvent remained in the receiving bottle, it was dried in a water bath, then dried at 100 °C ± 5 °C for 1 h, cooled in a dryer for 0.5 h, and weighed.

The fat content was calculated as follows:X = (m_1_ − m_0_)/m_2_ × 100(5)

X—Fat content in the dried chili pepper powder, mg/g;m_1_—Total weight of the receiving bottle and sample, g;m_0_—Mass of the receiving bottle, g;m_2_—Weight of the dried chili pepper, g.

### 2.11. HS-GC-TOF MS Analysis

GC–TOF MS was performed according to an improved version of the method of Muto et al. [18]. A 2 g sample was added to the headspace sample bottle with a rotor, shaken at 250 r·min^−1^ for 15 min, extracted with a 50/30 µm DVB/CAR/PDMS extractor at 50 °C for 30 min, and desorbed for 5 min.

Column: 30 m × 0.25 mm × 0.25 μm; carrier gas: He, with a constant flow rate of 1.0 mL/min and no shunt injection; inlet temperature: 260 °C. The heating procedure was as follows: the initial temperature was 40 °C for 5 min; then, the temperature was increased to 220 °C at a rate of 5 °C/min, increased to 250 °C at 20 °C/min, and held for 205 min. Interface temperature: 260 °C; ion source temperature: 230 °C; quadrupole rod temperature: 150 °C; ionization mode: EI, 70 eV; scanning mode: full scanning; *m*/*z* range: 20–400. Comparison was performed with the NIST2017 spectrum library.

### 2.12. Data Analysis

SPSS 26.0 statistical software was used to analyze the dry chili pepper data. When *p* < 0.05, significant differences were considered to exist between samples.

## 3. Results

### 3.1. Sensory Evaluation

Using color, taste, and fragrance as scoring indicators, the popularity of different varieties of dried chili peppers was analyzed through sensory evaluation, as shown in Figure 1. Among the dried chili pepper powders, the odor scores of CHX, CM, CN, CSW, CBR, and CYS were the highest. The color scores of CHN, CXZ, CG, CD, CX, CYS, XHR, XG, and XM were the highest. The taste scores of CGR, CBR, and XHR were the highest. CHX, CN, CBR, CYS, and XHR have higher combined scores.

### 3.2. Evaluation of Fruit Shape Index and Nutritional Index

The edible rate and fruit shape index of 18 types of dried chili pepper powder are presented in Table 1 and Appendix A. The edible rate of XM (72.94%) was the highest, followed by that of XHS (71.85%), CGR (71.38%), and CHX (69.26%), while the edible rate of CG (41.88%) was the lowest. CGR (8.69) had the highest fruit shape index, while CBR (8.68), CXZ (8.21), XM (6.43), and CD (1.35) had the lowest fruit shape indices. The fruit shape index of the same variety of chili pepper may vary due to differences in water content [19], as observed for CGR (8.69), CBR (8.68), CM (4.06), and CGM (4.45).

The nutritional indices of the 18 varieties of dried chili peppers are presented in Table 2 and Appendix A. The protein content of CHN was the highest (15.73 mg/g), followed by that of CX (15.70 mg/g), CM (15.13 mg/g), and CYS (14.65 mg/g), while the protein content of CD was the lowest (12.05 mg/g). CBR had the highest vitamin C content (3.88 mg/g), followed by CXZ (3.76 mg/g), CD (3.52 mg/g), and CGR (3.41 mg/g), and CX had the lowest vitamin C content (2.45 mg/g). The percentage of vitamins in chili peppers increases with increasing maturity, and hot air drying leads to a certain loss of vitamin C [20]. CGR had the highest soluble sugar content (26.06%), followed by that of CD (22.63%) and XHS (21.85%), and CYS had the lowest soluble sugar content (6.62%). CYS had the highest fat content (16.90 g/100 g), followed by CM (14.40 g/100 g) and CSS (13.70 g/100 g), and XHS had the lowest fat content (6.70 g/100 g).

### 3.3. HS-GC-TOF MS Analysis

The volatile components of dried chili peppers are relatively complex, and traditional one-dimensional chromatography cannot fully identify them. Therefore, HS-GC-TOF MS was used to extract and analyze the volatile components of different varieties of dried chili peppers, and a total of 380 volatile organic compounds were detected (Appendix A). These compounds included 62 alcohols, 50 aldehydes, 68 ketones, 60 hydrocarbons, 99 esters, 18 acids, and 23 other substances, including pyrazines and ethers. The percentage of volatile organic compounds was plotted (Figure 2) to directly describe the differences in the volatile components of different varieties of dried chili peppers.

As shown in the figure, there were significant differences in the contents of aldehydes, alkenes, and esters among the 18 dry chili peppers, and the contents of aldehydes in XHR (21.50%) and XHS (28.33%) were greater than those in the other varieties. The contents of alkenes in CHN (64.35%) and CGM (53.97%) were greater than those in the other chili peppers, and the contents of esters in CBR (14.60%), XG (19.10%) and CM (17.81%) were greater than those in the other chili peppers. In most dry chili peppers, the content of alkenes is significantly higher than that of other ingredients. Previous studies have shown that different varieties of chili pepper retain certain key complex aromas after drying; in this study, the complex aromas of different varieties differed, which is consistent with the literature [21].

Heatmap cluster analysis of alcohols in the volatile organic compounds of dried chili pepper was performed (Figure 3a). Alcohols are classified into saturated alcohols, unsaturated alcohols, and cyclic alcohols, of which saturated alcohols have a higher threshold of detection and are not easily detected [22]. The ethanol peak areas of CHN (5240.12), CYS (4651.96), and CSW (4097.41) were significantly higher than those of the other dry chili peppers, and CXZ (2844.68) had the lowest ethanol peak area. Linalool, the second most prevalent alcohol identified, is an aromatic monoterpene alcohol widely present in essential oils [23]. CXZ (4005.35), CHN (3577.81), and XM (2947.18) had the largest peak areas for linalool, and CGM (43) had the smallest peak area. 4-methyl-1-pentanol is the third largest alcohol substance, and the peak area content of CM (1414.34), CN (1173.64), and CSW (810.89) is significantly higher than that of other dried chili peppers; the volatile compound was also detected in the peel of Zi Yang tangerine [24]. n-Butyl alcohol is often used as a reference gas for complex gas mixtures [25]. Nerolidol, which can provide floral and fresh fruit aromas [23], was detected in all the samples, and the peak areas of XM (62.14) and CBR (61.45) were significantly higher than those of the other samples, consistent with the sensory score of 3.1.

Heatmap clustering analysis was performed for aldehydes in the volatile organic compounds of dried chili pepper (Figure 3b). The low aldehyde threshold plays a very important role in providing the characteristic smell of dried chili pepper. Isobutyraldehyde was the main volatile aldehyde; its peak areas in CYS (795.77), XR (781.70), and CM (713.02) were significantly greater than those in the other dried chili peppers, whereas the peak area of XG (89.37) was the lowest. 3-Methylbutanal was the second most prevalent volatile aldehyde, and its peak area in XHS (553.21) was significantly higher than that of the other dry chili pepper samples; the samples with the next highest peak areas were CG (388.37) and XHR (365.26), and CHN (68.12) had the lowest peak area. Propionaldehyde was the third most prevalent volatile aldehyde; CYS (740.67) had the largest peak area, followed by CHN (642.29) and XHR (442.53), but no propionaldehyde was detected in CSW, CXZ, XM, CN, CSS or CD. Furfural has a burnt and malty taste and is one of the main flavor substances in malt [26]. Among the tested chili pepper varieties, higher peak areas for this compound were detected for CGR (217.03) and CG (215.60). 2,4-Heptadienal is a volatile aldehyde with a high content in chili oil and can provide a grassy and fruity flavor [27]. In this study, the peak area of 2,4-heptadienal in XHS (147.02) was high, but this compound was not the main volatile aldehyde, possibly because the bio-oil used in the production of chili oil is not affected by temperature and contains mainly aldehydes and ketones [28]. trans-2-Pentenal, 1,3,4-trimethyl-3-cyclohexen-1-carboxaldehyde, and 3,3-diethoxypropyne were detected only in a few samples and had a small peak area.

The ketones in the volatile organic compounds of dried chili pepper were analyzed. Volatile ketones are also important components of volatile organic compounds, but ketones are typically not the main odorants. The 3-octanone peak areas of CHN (265.09), CYS (205.16), and CBR (204.32) were the largest. 2-Octanone, acetol, 3-hepten-2-one, 4-hydroxy-4-methyl-5-hexenoic acid gamma lactone, 1-methyl-2-pyrrolidinone, ketoisophorone, p-methyl acetophenone, dihydro-beta-ionone, alpha-ionone, 2,3-dihydro-3,5-dihydroxy-6-methyl-4h-pyran-4-one, and 3-ethyl-4-methyl-1h-pyrrole-2,5-dione are volatile ketones that are prevalent in all varieties of chili peppers. Beta-ionone is mainly attributed to the degradation of carotenoids [29]. The peak area of beta-ionone was the largest in CHN (37.87), followed by XG (19.01) and CN (22.03), and XM (4.00) had the lowest peak area. Ketones accounted for a relatively low proportion of the volatile organic compounds in the dried chili pepper samples.

Heatmap cluster analysis of hydrocarbons in the volatile organic compounds of dried chili pepper was carried out (Figure 3c). Previous studies have shown that the characteristic odor substances in oven-dried chili pepper powder are mainly aldehydes and terpenes [30]. Niu et al.’s [31] study has shown that laurene, 1-caryophyllene, trans-alpha-basil, and limonene can cause differences in the flavor of chili oil. In this study, volatile hydrocarbons were present in different varieties of dried chili peppers. CHN had the largest peak area for limonene (1046.51), and the overall sensory score of CHN was also relatively high. CYS (4902.50) and CM (4774.81) presented the next highest sensory scores, and CM had high odor scores in the sensory evaluation. Limonene was not detected in CGM, CSS, CD, or CG. 1,2,4,4-Tetramethylcyclopentene was the second most prevalent volatile hydrocarbon in the dried chili pepper samples, with CHX (3747.53) having the largest peak area, followed by CHN (2565.14) and CYS (2354.39). No volatile hydrocarbons were detected in CM, CGR, XM, CSS, or CG. The volatile hydrocarbons alpha-phellandrene, myrcene, sabinene, and beta-ocimene were present in all the dried chili peppers.

Analysis of esters in the volatile organic compounds of the dried chili pepper samples revealed that due to their low threshold, esters generally contributed strongly to the odor of the samples (Figure 3d). Volatile esters were also the main volatile organic compounds identified in this study. n-Butyl acetate, n-hexyl acetate, hexyl 2-methylbutyrate, hexenyl valerate, (3z)-, methyl salicylate, delta-hexalactone, tri-isobutylphosphate, pantolactone, pantolactone, ethyl cinnamate, 2-hydroxy-gamma-butyrolactone, dihydroactinidiolide, diisobutyl phthalate, and dibutyl phthalate are common volatile esters in all dried chili peppers. Some studies showed that the content of volatile esters in chili pepper increased during fermentation, and ethyl palmitate and ethyl linoleate were the main esters [28]. In this study, ethyl caprylate was the ethyl ester with the highest peak area in CX (2696.15), CM (3373.63), and CBR (2693.23) and was the main volatile ester in these samples. However, ethyl caprylate was not detected in CSW, XHS, CN, CGN, CX, or CG. Methyl acetate was the second most prevalent ester, with the largest peak area in CHN (544.62), followed by CSW (527.48) and CYS (458.53). The content of esters is usually affected by the capsaicin content in chili pepper [32].

The acids in the volatile organic compounds of dried chili pepper were analyzed. Acetic acid, propionic acid, isobutyric acid, 2-methylbutanoic acid, isocaproic acid, and hexanoic acid are volatile acids commonly found in dried chili peppers. Acetic acid is the most active volatile acid. CGR (4587.35) had the largest peak area for acetic acid, followed by CSS (4072.65) and CSW (4171.57), and CN (770.78) had the smallest peak area. Propionic acid was the second most prevalent volatile acid, with XHS (446.59) having the largest peak area, followed by CM (348.00) and XG (227.63), and XM (69.24) had the smallest peak area.

Other substances among the volatile organic compounds in dried chili pepper, mainly including pyrazines and ethers, were analyzed. CYS and XM contained the highest contents of other volatile organic compounds, among which pyrazines and ethers were present in all the dried chili pepper varieties. Pyrazines provide nutty and barbecue flavors and are important aroma components [27]. 2,5-Dimethylpyrazine was the main volatile pyrazine and had the largest peak area in CHN (100.98), followed by XHR (53.07) and XG (67.72), but it was not detected in CGM, CSS, or CG. Diethyl ether was the main volatile ether; CYS (3287.09) had the largest peak area, followed by CHN (3259.03) and CN (2348.55), and XHS (151.55) had the smallest peak area.

## 4. Discussion

It can be seen from the sensory review results of Section 3.1 that the volatile flavor of dried pepper is positively correlated with the sensory score, and the overall sensory score of dried pepper with good volatile flavor is high. In addition, the presence of volatile esters was positively correlated with the aroma of dried chili peppers, indicating that the difference in the aroma of dried chili peppers might be related to the type and content of volatile esters in different varieties [27], which was consistent with the result of Section 3.3. In this study, several dried chili peppers with higher volatile ester content also had higher overall sensory scores. The overall sensory score of dried peppers is affected by the drying temperature. Some studies have reported that when the drying temperature is greater than 60 °C, the loss of volatile components in dried chili powder increases [33], and the sensory score of dried chili pepper will be different due to the loss of volatile components.

There are many kinds of chili peppers, and different regions have different classifications and naming conventions. The quality of chili is mainly determined by its taste, nutritional index, and fruit shape index. The same variety can also be affected by local variation, resulting in differences in fruit shape index. In addition, the application of chili varies according to variety and fruit shape index [34]. Dried peppers with a larger fruit index are usually made into paprika, while peppers with a smaller fruit index are usually used whole. The consumption rate of dried pepper is low, and XM is a spicy variety of pepper, mainly used for fresh consumption, so the consumption rate is high; CG is mainly used in hot pot to provide spicy flavor and is not eaten directly, and the consumption rate is low. In addition to influencing consumers’ choice of chili peppers, the nutritional index also affects the flavor of chili peppers. Protein is involved in the synthesis and storage of carotenoids in capsicum [35], which may have a certain impact on the color of capsicum. Carotenoids are also one of the precursors of pepper aroma components, and the cracking reaction of carotenoids will produce geranyl aldehyde, β-ionone, and other important volatile components in dried pepper [36]. According to the sensory evaluation results in Section 3.1, the high color scores of CHN and CX may be related to their high protein content.

In this work, HS-GC-TOF MS analysis of the volatile organic compounds in 18 kinds of dried chili pepper was carried out. The contents of volatile organic compounds in different samples were determined by peak area normalization. Ethanol, hexaldehyde, 1,2,4,4-tetramethylcyclopentene, limonene, beta-ocimene, ethyl caprylate, acetic acid, and diethyl ether were the volatile organic compounds with the highest contents in the 18 kinds of dried chili peppers.

In CHN, CX, CHX, CXZ, XM, CM, and CD, olefins account for more than 20% of the total VOCs. Studies have shown that the secondary structure of proteins and proteins can affect alkanes and olefins, thus affecting the flavor characteristics of meat analogs [37]. In this study, the high protein content of CX, CM, and CHX may be one of the reasons for the highly volatile olefin content of CX, CM, and CHX. At the same time, the high protein content of peppers will affect the synthesis of terpene in peppers, and the key enzyme in the synthesis of terpene is the main reason for the formation of the fruit flavor of peppers [38]. These studies have shown that dried peppers with higher protein content also have higher levels of volatile components. CD, CHX, and CX have higher soluble sugar content. Previous studies have found that the distribution of volatile components in durian during drying is positively correlated with soluble sugar content [39], and the distribution of volatile components in lemon during drying is also correlated with soluble sugar content [40], indicating that the distribution of volatile components in dried pepper may also be related to its soluble sugar content. Gao et al. [41] reported correlations between volatile and non-volatile components of *Poria cocos*. In summary, there is also a certain correlation between volatile and non-volatile components of dried pepper, which jointly affect the sensory score of dried pepper.

## 5. Conclusions

The sensory scores, non-volatile components, and volatile components of dried chili peppers were analyzed. The results showed that the sensory scores of the dried chili peppers were strongly affected by their color and odor. There were differences in the contents of non-volatile components, among which CYS, CGR, CD, and CM had higher contents. The contents of total volatile components of CX, CSS, CM, CHN, and XG were high, and the volatile components varied greatly among the different varieties. The types and contents of volatile components in dried chili peppers with higher contents of non-volatile components were also higher, which further affected the sensory scores of the dried chili peppers. The results are helpful for selecting the appropriate raw materials for use in different chili pepper products.

## Figures and Tables

**Figure 1 foods-14-00712-f001:**
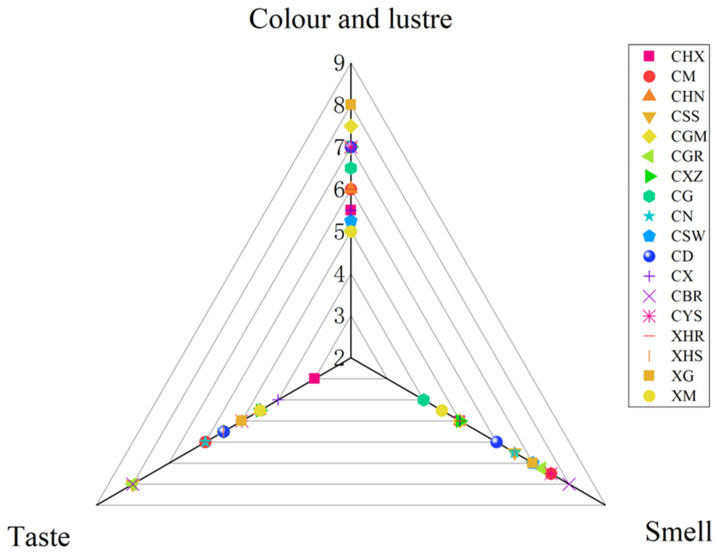
Dried chili pepper sensory score.

**Figure 2 foods-14-00712-f002:**
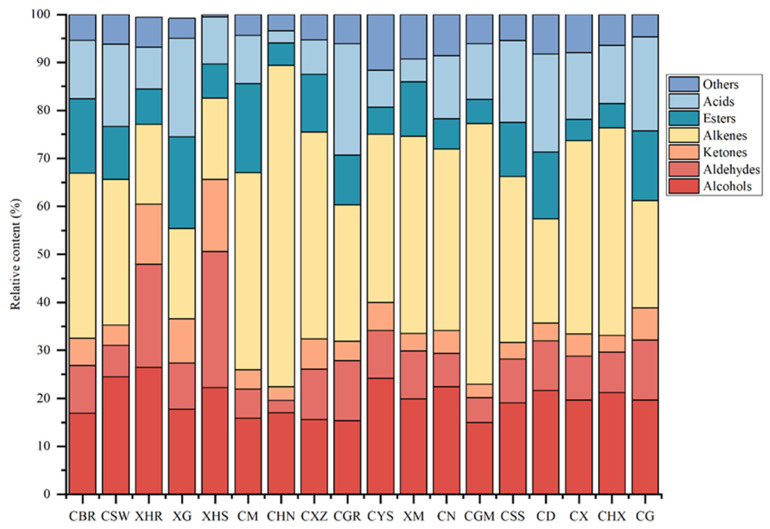
Stack diagram of volatile organic compounds in different dried capsicums based on GC-TOF MS.

**Figure 3 foods-14-00712-f003:**
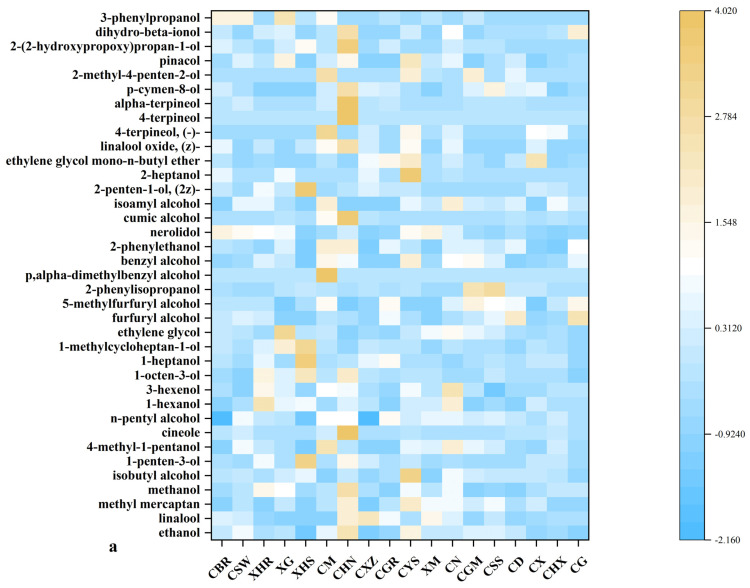
Volatile organic compound heat maps of different dried chili powders: (**a**) volatile alcohol heat map of dried chili powder; (**b**) volatile aldehyde heat map of dried chili powder; (**c**) volatile hydrocarbon heat map of dried chili powder; (**d**) volatile ester heat map of dried chili powder.

**Table 1 foods-14-00712-t001:** Edible rate and fruit type index of dried capsicum.

Sample	Weight of Single Fruit (g)	Remove Stalk and Seed Weight (g)	Fruit Length (cm)	Fruit Width (cm)	Fruit Type Index	Edible Rate (%)
CHX	1.07 ± 0.22 ^efg^	0.73 ± 0.16 ^de^	5.63 ± 0.68 ^gh^	1.62 ± 0.19 ^ef^	3.49 ± 0.36 ^e^	69.26 ± 10.25 ^ab^
CM	0.44 ± 0.11 ^i^	0.20 ± 0.06 ^h^	4.69 ± 0.68 ^hij^	1.16 ± 0.13 ^ij^	4.06 ± 0.50 ^de^	43.94 ± 4.35 ^hi^
CHN	0.46 ± 0.10 ^i^	0.26 ± 0.04 ^gh^	4.09 ± 0.64 ^j^	1.17 ± 0.12 ^hij^	3.48 ± 0.33 ^e^	58.32 ± 7.72 ^de^
CSS	0.98 ± 0.17 ^efgh^	0.45 ± 0.06 ^fgh^	5.56 ± 0.52 ^gh^	1.56 ± 0.24 ^efg^	3.59 ± 0.36 ^e^	46.52 ± 5.02 ^ghi^
CGM	0.74 ± 0.09 ^hi^	0.42 ± 0.07 ^fgh^	5.99 ± 0.35 ^fg^	1.35 ± 0.08 ^ghi^	4.45 ± 0.44 ^cd^	57.44 ± 7.50 ^de^
CGR	1.97 ± 0.27 ^d^	1.40 ± 0.20 ^c^	14.57 ± 1.79 ^b^	1.68 ± 0.12 ^de^	8.69 ± 1.06 ^a^	71.38 ± 6.10 ^ab^
CXZ	1.31 ± 0.22 ^e^	0.85 ± 0.14 ^d^	10.73 ± 2.08 ^c^	1.41 ± 0.18 ^fgh^	8.21 ± 1.05 ^a^	65.79 ± 8.62 ^bc^
CG	1.28 ± 0.33 ^ef^	0.53 ± 0.13 ^ef^	3.93 ± 0.31 ^j^	1.75 ± 0.17 ^de^	2.25 ± 0.18 ^f^	41.88 ± 4.99 ^i^
CN	0.64 ± 0.20 ^hi^	0.33 ± 0.09 ^fgh^	4.98 ± 0.73 ^ghij^	1.24 ± 0.10 ^hi^	4.04 ± 0.59 ^de^	51.68 ± 6.47 ^efg^
CSW	0.88 ± 0.26 ^fgh^	0.49 ± 0.13 ^eg^	6.72 ± 0.98 ^f^	1.35 ± 0.14 ^ghi^	4.98 ± 0.65 ^c^	56.52 ± 7.80 ^def^
CD	3.21 ± 0.48 ^b^	1.58 ± 0.21 ^c^	4.17 ± 0.61 ^ij^	3.14 ± 0.37 ^a^	1.35 ± 0.28 ^g^	49.83 ± 6.74 ^fgh^
CX	0.90 ± 0.17 ^fgh^	0.54 ± 0.10 ^ef^	5.18 ± 0.95 ^ghi^	1.34 ± 0.18 ^ghi^	3.91 ± 0.70 ^de^	60.41 ± 8.51 ^cd^
CBR	2.95 ± 0.41 ^bc^	2.02 ± 0.35 ^b^	15.78 ± 1.54 ^a^	1.87 ± 0.37 ^d^	8.68 ± 1.60 ^a^	68.15 ± 3.23 ^ab^
CYS	0.65 ± 0.15 ^hi^	0.33 ± 0.10 ^fgh^	7.86 ± 1.02 ^e^	1.11 ± 0.08 ^ij^	7.07 ± 0.81 ^b^	50.63 ± 7.38 ^fg^
XHR	2.86 ± 0.70 ^bc^	1.55 ± 0.34 ^c^	5.78 ± 0.76 ^fg^	3.12 ± 0.44 ^ab^	1.88 ± 0.32 f^g^	54.79 ± 7.46 ^def^
XHS	3.76 ± 0.94 ^a^	2.69 ± 0.66 ^a^	12.28 ± 1.92 ^c^	2.91 ± 0.39 ^bc^	4.24 ± 0.55 ^de^	71.85 ± 5.10 ^ab^
XG	2.84 ± 0.78 ^bc^	1.64 ± 0.42 ^c^	9.58 ± 0.86 ^d^	2.78 ± 0.32 ^c^	3.48 ± 0.42 ^e^	58.60 ± 5.39 ^d^
XM	2.63 ± 0.51 ^c^	1.91 ± 0.38 ^b^	5.95 ± 0.58 ^fg^	0.98 ± 0.25 ^j^	6.43 ± 1.61 ^b^	72.94 ± 6.57 ^a^
Max	3.76	2.69	15.78	3.14	8.69	72.94
Min	0.44	0.20	3.93	0.98	1.35	41.88
range	3.32	2.49	11.85	2.16	7.34	31.06
Standard deviation	1.10	0.75	3.65	0.72	2.24	9.86
Mean value	1.64	1.00	7.42	1.75	4.68	58.33
Coefficient of variation (%)	66.75	74.87	49.25	41.15	47.86	16.90

Different lowercase letters of shoulder labels in the same column indicated significant differences (*p* < 0.05).

**Table 2 foods-14-00712-t002:** Nutritional indexes of dried capsicum.

Sample	Ash Content (g/100 g)	Protein (mg/g)	Vc (mg/g)	Soluble Sugar (%)	Fat Content (g/100 g)
CHX	9.03 ± 0.11 ^de^	13.99 ± 0.04 ^def^	3.06 ± 0.11 ^fgh^	16.08 ± 0.01 ^g^	9.80 ± 0.17 ^gh^
CM	6.38 ± 0.03 ^i^	15.13 ± 0.09 ^b^	3.13 ± 0.01 ^defg^	12.84 ± 0.19 ^k^	14.40 ± 0.13 ^b^
CHN	9.04 ± 0.01 ^de^	15.73 ± 0.17 ^a^	3.21 ± 0.08 ^cdefg^	9.27 ± 0.02 ^o^	10.70 ± 0.03 ^efg^
CSS	8.00 ± 0.17 ^fg^	13.15 ± 0.03 ^gh^	2.97 ± 0.03 ^gh^	14.59 ± 0.08 ^j^	13.70 ± 0.02 ^bc^
CGM	8.12 ± 0.19 ^efg^	14.40 ± 0.21 ^cd^	2.83 ± 0.02 ^hi^	17.97 ± 0.14 ^e^	12.30 ± 0.07 ^cde^
CGR	6.82 ± 0.05 ^hi^	12.26 ± 0.07 ^ij^	3.41 ± 0.13 ^bc^	26.06 ± 0.05 ^a^	8.70 ± 0.07 ^hi^
CXZ	10.80 ± 0.21 ^c^	13.38 ± 0.30 ^g^	3.76 ± 0.36 ^a^	17.50 ± 0.01 ^f^	8.70 ± 0.05 ^hi^
CG	8.99 ± 0.13 ^de^	14.58 ± 0.14 ^bcd^	3.13 ± 0.05 ^defg^	12.60 ± 0.10 ^l^	12.90 ± 0.01 ^bcd^
CN	10.80 ± 0.03 ^c^	13.48 ± 0.02 ^fg^	3.38 ± 0.20 ^bcd^	14.49 ± 0.18 ^j^	12.00 ± 0.12 ^def^
CSW	7.73 ± 0.31 ^fgh^	13.61 ± 0.01 ^efg^	3.11 ± 0.01 ^efg^	16.05 ± 0.01 ^g^	11.20 ± 0.03 ^efg^
CD	8.40 ± 0.02 ^ef^	12.05 ± 0.05 ^j^	3.52 ± 0.02 ^b^	22.63 ± 0.13 ^b^	11.80 ± 0.13 ^def^
CX	9.63 ± 0.05 ^d^	15.70 ± 0.23 ^a^	2.45 ± 0.09 ^j^	18.53 ± 0.08 ^d^	7.85 ± 0.30 ^ij^
CBR	11.40 ± 0.12 ^bc^	12.66 ± 0.09 ^hi^	3.88 ± 0.20 ^a^	12.42 ± 0.21 ^m^	8.90 ± 0.13 ^hi^
CYS	7.6 ± 0.02 ^fgh^	14.65 ± 0.07 ^bc^	2.65 ± 0.12 ^ij^	6.62 ± 0.06 ^p^	16.90 ± 0.01 ^a^
XHR	11.60 ± 0.03 ^bc^	14.20 ± 0.11 ^cde^	3.26 ± 0.05 ^cdef^	11.04 ± 0.05 ^n^	9.65 ± 0.17 ^gh^
XHS	12.20 ± 0.05 ^b^	12.11 ± 0.02 ^ij^	2.67 ± 0.23 ^ij^	21.85 ± 0.25 ^c^	6.70 ± 0.03 ^j^
XG	14.10 ± 0.03 ^a^	14.05 ± 0.01 ^cdef^	2.83 ± 0.06 ^hi^	14.82 ± 0.23 ^i^	10.60 ± 0.05 ^fg^
XM	7.37 ± 0.01 ^gh^	13.04 ± 0.03 ^gh^	3.32 ± 0.02 ^bcde^	15.70 ± 0.02 ^h^	9.65 ± 0.14 ^gh^
Max	14.10	15.73	3.88	26.06	16.90
Min	6.38	12.05	2.45	6.62	6.70
range	7.72	3.68	1.43	19.44	10.20
Standard deviation	2.08	1.14	0.38	4.77	2.53
Mean value	9.33	13.79	3.14	15.61	10.91
Coefficient of variation (%)	22.30	8.24	12.02	30.57	23.23

Different lowercase letters of shoulder labels in the same column indicated significant differences (*p* < 0.05).

## Data Availability

The original contributions presented in this study are included in the article/Appendix A. Further inquiries can be directed to the corresponding author.

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
