# Peer review of "Analysis of Volatile and Non-Volatile Components of Dried Chili Pepper (Capsicum annuum L.)"

_foods, 2025, doi:10.3390/foods14050712_

Round 1
Reviewer 1 Report
Comments and Suggestions for Authors
The manuscript contains quite interesting research results with varieties of chilli peppers. Why did the Authors tackle such a research topic? I would like to ask the Authors to clearly indicate the economic applicability of the research results obtained. What, if any, is an obstacle to other implementations? What should now be further investigated? To clarify?
Materials and Methods
Where was the research carried out? In which year?
How were people selected for testing? Was the selection completely random? Does gender matter?
Please provide full details of the producer of the statistical software used to analyse the data.
Discussion
It is very short, please expand on it.
References
Please limit to publications that were published in 2015 or later.
Reviewer 2 Report
Comments and Suggestions for Authors
foods-3456206
Analysis of volatile and non-volatile components of dried chili pepper
1- Abstract
Express the purpose of the paper more clearly.
2- Abstract
Mention the methods used or according to which standards they are They only mention HS-GC-TPF MS, for which they should define what this acronym means.
3- 2.1 Materials
Lines 65-66
“The dried chili peppers were purchased from Xinjiang, Sichuan and Chong
qing, China.”.
Mention what type of company it is
4- Lines 71- 73
“ After all the chili peppers were dried by hot air at different temperatures, they were removed from the stalks, crushed with a grinder and passed through a 40-mesh sieve. The chili pepper powder was stored in a refrigerator at -20 ℃.”
Is this process carried out by the company or is it a step after the purchase of the samples? It is confusing. Additionally, include information on temperatures, which you mention
5- Discussion
Lines 381-383
Specifically mention the most relevant volatile esters.
6- Please expand and improve your presentation, be more specific, mention the most relevant results.
7- Table S1Volatile organic compounds peak area measured by GC-TOFMS.
Improve the title, indicating what type of samples they are
Include in the first row, if appropriate: "Chili pepper varieties"
After the suggested changes, the paper should be accepted for publication.
Reviewer 3 Report
Comments and Suggestions for Authors
Title
Include the name of the genus and species of the chili pepper (Capsicum annuum L.). “Analysis of volatile and non-volatile components of dried chili pepper (Capsicum annuum L.)”
Introduction
Include in the Background the characteristics of other species of the Capsicum genus, for example C. chinense Jacq., known as Habanero chili, which is highly valued for its sensory attributes such as color, smell and pungency. The characteristic aroma of the Habanero chili is provided by some of the volatile compounds present in the fruit, see: Murakami Y, Iwabuchi H, Ohba Y, Fukami H. Analysis of Volatile Compounds from Chili Peppers and Characterization of Habanero (Capsicum chinense) Volatiles. J Oleo Sci. 2019;68(12):1251-1260. doi: 10.5650/jos.ess19155
Material and Methods
Plant material. Specify that the study material is chili pepper (Capsicum annuum L.), because there are other species of chili with the same name.
Results and Discussion
The authors state that not all volatile components of a food contribute significantly to aroma, and only a few key volatile components contribute to the overall flavor. Based on the experimental data obtained in the present study, what would be the key volatile components that contribute to aroma in the studied chili fruits.
Reviewer 4 Report
Comments and Suggestions for Authors
The paper "Analysis of volatile and non-volatile components of dried chili pepper" addresses an interesting topic concerning the differences between various chilli pepper varieties. The differences detected by the authors may contribute to a better utilization of specific varieties. Therefore, the study has practical significance, and the information provided can be useful not only from a scientific perspective but also for the food industry. However, despite the merits of this manuscript, revisions are necessary before publication. My comments are presented below.
1. The introduction should be revised. It focuses on describing research tools and their obvious applications (e.g., GC analysis for studying volatile compounds). Instead, the introduction should at least briefly outline what is already known in the field of the study (e.g. in the study of chilli pepper), providing background for the undertaken research. Please modify the introduction to ensure it provides an appropriate background.
2. The authors state in section 2.2 the Preparation of chilli pepper: "After all the chilli peppers were dried by hot air at different temperatures." It would be beneficial to include a table (e.g., in the supplementary materials) specifying the exact drying temperature for each chilli pepper variety.
3. The heat maps are illegible (the names of the compounds need to be larger or of better quality).
4. In Figures 2 and 3, the SD bars are missing. Additionally, the figures repeat information already presented in the tables. According to the general principle, data should be presented in one selected format—either tables or figures. Therefore, I suggest moving one of these formats to the supplementary materials. Moreover, Figures 2 and 3 have both a primary and secondary Y-axis with different units. However, the legend does not specify which variable is expressed in which unit, making it difficult to interpret the graphs. Readers have to refer back to the table. This information should be included in the legend or caption.
5. Though Figure 1 is interesting, the colours representing the varieties are sometimes too similar (e.g., CX and CBR or CHX and CYS). Adjustments are needed—perhaps instead of points, the authors could use x markers to differentiate the lines in similar colours.
6. The discussion is somewhat chaotic and lacks a coherent narrative. The authors need to make revisions to ensure logical structure.
7. The discussion ends with a summary paragraph, which is unnecessary. The authors have already provided a Conclusion for summarization.
Round 2
Reviewer 4 Report
Comments and Suggestions for Authors
I would like to thank the Authors for the changes they have made, which have improved the quality of the manuscript. I believe that the article can be published in this version.